# New Mechanisms of Bromelain in Alleviating Non-Alcoholic Fatty Liver Disease-Induced Deregulation of Blood Coagulation

**DOI:** 10.3390/nu14112329

**Published:** 2022-06-01

**Authors:** Po-An Hu, Sz-Han Wang, Chia-Hui Chen, Bei-Chia Guo, Jenq-Wen Huang, Tzong-Shyuan Lee

**Affiliations:** 1Department and Graduate Institute of Physiology, College of Medicine, National Taiwan University, Taipei 10051, Taiwan; d07441003@ntu.edu.tw (P.-A.H.); hjg231020@gmail.com (S.-H.W.); chiahuichen1993@ntu.edu.tw (C.-H.C.); d07441002@ntu.edu.tw (B.-C.G.); 2Department of Internal Medicine, National Taiwan University Hospital, Taipei 10051, Taiwan; 3Department of Internal Medicine, College of Medicine, National Taiwan University, Taipei 10051, Taiwan

**Keywords:** bromelain, non-alcoholic fatty liver disease, blood coagulation, intrinsic pathway, extrinsic pathway, fibrinolytic system

## Abstract

Bromelain, an enzyme extracted from the stems of pineapples, exerts anticoagulant effects; however, the regulatory mechanisms are not fully understood. Here, we aimed to investigate the effects of bromelain on non-alcoholic fatty liver disease (NAFLD)-induced deregulation of blood coagulation and the underlying molecular mechanisms. C57BL/6 mice were fed a high-fat diet (HFD), with or without bromelain (20 mg/kg/day) administration, for 12 weeks. Treatment with bromelain decreased thrombus formation in the liver and prolonged HFD-induced shortened prothrombin, activated partial thromboplastin, and fibrinogen times. Moreover, liquid chromatography-mass spectrometry/mass spectrometry and Western blot analysis showed that bromelain inhibited NAFLD-induced activation of the intrinsic, extrinsic, and common pathways by upregulating the protein expression of antithrombin III, serpin family G member 1, and α1-antitrypsin, and downregulating the protein expression of fibrinogen in the liver and plasma. Bromelain also upregulated the level of plasminogen and downregulating factor XIII expression in the liver and plasma. Collectively, these findings suggest that bromelain exerts anticoagulant effects on NAFLD-induced deregulation of coagulation by inhibiting the activation of the coagulation cascade, decreasing the stability of clots, and promoting fibrinolytic activity. The present study provides new insights into the potential therapeutic value of bromelain for the prevention and treatment of thrombosis-related diseases.

## 1. Introduction

Bromelain is a cysteine-rich protease found in the stem, fruit, leaves, and peel of pineapples. Stem bromelain [EC 3.4.22.32] has been used as a drug for treating acute inflammation and inflammation-associated diseases, such as osteoarthritis, and improving postoperative pain and swelling [1,2,3]. In addition to its anti-inflammatory properties, bromelain has potent anticoagulant properties [4,5]; bromelain prolongs prothrombin time (PT) and activated partial thromboplastin time (APTT) by inhibiting the activation of intrinsic and extrinsic pathways, thereby preventing clot formation [6]. Additionally, bromelain prolongs blood clotting time by increasing fibrin cleavage without degrading fibrinogen (FIB) [7]. Given its anti-thrombotic and fibrinolytic properties, bromelain has been recommended for the prevention of cardiovascular events, such as heart attack and stroke [8]. Although bromelain is considered a potential drug for the treatment of coagulation disorders, the underlying molecular mechanisms that regulate its anticoagulant activity have not been fully elucidated.

The popularity of the Western diet worldwide has led to the rise of metabolic syndrome-related diseases, including obesity, hypertension, diabetes, and atherosclerosis, which have become global health issues [9,10]. Several lines of evidence suggest that obesity is strongly associated with venous thromboembolic disease and arterial thrombosis; therefore, obesity is considered a risk factor for thrombosis [11,12]. Production of coagulation factors is one of the main physiological functions of the liver; hence, the effect of liver diseases, such as non-alcoholic fatty liver disease (NAFLD), on coagulation has been well documented [13]. Obesity-induced NAFLD is associated with increased expression of factor VIII and decreased expression of protein C, which leads to the aberrant activation of the coagulation system and the formation of blood clots [14]. Moreover, the activities of factors VIII, IX, XI, and XII are increased in patients with NAFLD [15]. These observations suggest that NAFLD may cause overactivation of the intrinsic and extrinsic pathways to enhance thrombus formation [15]. Furthermore, NAFLD is associated with deregulation of the fibrinolytic system; the level of plasminogen activator inhibitor-1 (PAI-1) is increased in patients with NAFLD, which results in inhibition of the fibrinolytic system, leading to fibrin accumulation and increased thrombus formation [16,17,18]. However, little is known about the molecular mechanisms underlying NAFLD-induced deregulation of the coagulation system. The liver is the main organ that produces clotting factors and anticoagulants; therefore, liver diseases, such as NAFLD, can induce thrombosis [17,18]. Therefore, it was necessary to establish a thrombosis model in our study and HFD-fed mice were used to induce NAFLD and thrombosis, and delineate the mechanisms underlying the effects of bromelain on NAFLD-induced deregulation of blood coagulation.

Given the effect of bromelain on the coagulation system, we aimed to explore its effects and underlying molecular mechanisms on high-fat diet (HFD)-induced dysregulation of the coagulation system in the blood and liver of mice. First, we investigated the effects of bromelain on PT, APTT, and FIB time, and thrombus formation in the liver of HFD-fed mice. Second, we used liquid chromatography-mass spectrometry/mass spectrometry (LC-MS/MS) to analyze the protein expression of coagulation factors in the liver and assess the potential molecular mechanism regulating the effects of bromelain on the coagulation system.

## 2. Materials and Methods

### 2.1. Reagents and Antibodies

Bromelain was purchased from Cayman Chemical Co. (Ann Arbor, MI, USA). Mouse antibodies against antithrombin III (ATIII, sc-271987) were purchased from Santa Cruz Biotechnology (Santa Cruz, CA, USA). Mouse antibodies against β-actin (AC004) were purchased from ABclonal Science (Woburn, MA, USA). Mouse antibodies against factor XIIIa (ab1834) and rabbit antibodies against plasminogen (ab154560) were purchased from Abcam (Cambridge, UK). Rabbit antibodies against serpin family G member 1 (SERPING1, 12259-1-AP) and α1-antitrypsin (16382-1-AP) were purchased from Proteintech (Rosemont, IL, USA). Rabbit antibodies against FIB α (bs-7548R) and FIB γ (bs-6895R-TR) were purchased from Bioss Inc. (Woburn, MA, USA). Rabbit antibodies against FIB β (ARG58574) were purchased from Arigo Biolaboratories (Hsinchu, Taiwan). PT, APTT, and FIB assay kits were purchased from TECO Medical Instruments (Neufahrn, Niederbayern, Germany).

### 2.2. Experimental Animals

The present study followed the Guide for the Care and Use of Laboratory Animals (Institute of Laboratory Animal Resources, eighth edition, 2011). All experiments involving animals were approved by the Animal Care and Utilization Committee of the National Yang-Ming University (Taipei, Taiwan) (No. 1070314). Eight-week-old male wild-type C57BL/6 mice were obtained from the National Laboratory Animal Center (Taipei, Taiwan). Mice were divided into two groups (n = 10 each group), fed an HFD (60% fat, Research Diets, New Brunswick, NJ, USA), and orally treated with phosphate-buffered saline or bromelain (20 mg/kg/day) for 12 weeks. Mice were placed in barrier facilities and maintained at a constant temperature (22 °C) and humidity (40–60%) with a 12 h light–dark cycle. At the end of the experiment, the mice were sacrificed using CO_2_ and the liver and blood were collected. These samples were then subjected to different analyses or preserved at −80 °C. The frozen liver samples were homogenized and blood samples were centrifuged to obtain plasma, which were then subjected to Western blot analysis.

### 2.3. Histological Examination

Liver specimens were cut into 8 μm sections and subjected to histological analysis. The deparaffinized sections were rehydrated, stained with hematoxylin and eosin Y (H&E stain), and observed under a Motic Panthera U microscope (Motic Images, Xiamen, China).

### 2.4. Anticoagulant Activity

The PT (A0230-010), APTT (A0300-025), and FIB (A0501-010) reagents and Coatron M1 Coagulation Analyzer were purchased from TECO Medical Instruments. For the PT assay, plasma was mixed with the PT reagent and incubated at 37 °C for 2 min, after which the clotting time was recorded. For APTT assay, plasma was mixed with APTT reagent and incubated at 37 °C for 3 min, after which CaCl_2_ was added, and the clotting time was recorded. For the FIB assay, plasma was mixed with FIB reagent, and the clotting time and FIB concentration were recorded.

### 2.5. LC-MS/MS

The samples were subjected to sodium dodecyl-sulfate polyacrylamide gel electrophoresis (SDS-PAGE), and the gel lanes were cut and digested. Peptides were separated using an Ultimate 3000 nanoLC system (Dionex LC-Packings, Amsterdam, The Netherlands) equipped with a 20 cm × 75 μm internal diameter (i.d.) fused silica column custom packed with 3 μm 120 Å ReproSil Pur C18 aqua (Dr. Maisch GMBH, Ammerbuch-Entringen, Germany). After injection, the peptides were trapped at 30 μL/min on a 5 mm × 300 μm i.d. Pepmap C18 cartridge (Dionex LC-Packings) and separated. Eluting peptides were ionized at 1.7 kV in a Nanomate Triversa Chip-based nanospray source using a Triversa LC coupler (Advion, Ithaca, NJ, USA). Intact peptide mass and fragmentation spectra were acquired on a LTQFT hybrid mass spectrometer (Thermo Fisher, Bremen, Germany). The Thermo raw files from tryptic digests were loaded into Progenesis QI proteomics software version 4.1 (NonLinear Dynamics, Northumberland, UK). Several criteria were used to filter the data before exporting the MS/MS output files to MASCOT (http://www.matriscience.com, accessed on 24 September 2017) for protein identification. All MS/MS spectra were exported from the Progenesis software as a MASCOT generic file (mgf) and used for peptide identification with MASCOT (version 2.6) searched against Mus musculus (house mouse) SwissProt 2017_07 (555,100 sequences; 198,754,198 residues). LC/MS/MS data were then uploaded from a Microsoft Excel spreadsheet onto FunRich 3.0 software (La Trobe University, Bundoora, Australia) (http://www.funrich.org, accessed on 11 October 2017) and Metacore 6.13 software (Clarivate, London, UK) (http://www.genego.com, accessed on 11 October 2017).

### 2.6. Western Blot Analysis

Liver samples were lysed with immunoprecipitation lysis buffer (50 mM Tris, pH 7.5, 5 mM EDTA, 300 mM NaCl, 1% Triton X100, 1 mM phenylmethylsulfonyl fluoride, 10 mg/mL leupeptin, and 10 mg/mL aprotinin and phosphatase inhibitor cocktails I and II). Aliquots of liver lysates (50 μg protein) were mixed with the loading dye (250 mM Tris HCl pH 6.8, 500 mM dithiothreitol, 10% SDS, 50% glycerol, and bromophenol blue) and denatured at 100 °C for 5 min. The plasma sample (1.5 μL) was mixed with immunoprecipitation lysis buffer and loading dye to bring the total volume up to 10 μL. Proteins were separated using 8% or 10% SDS-PAGE, and transferred onto a polyvinylidene difluoride (PVDF) membranes (Pall, Port Washington, WI, USA). The membranes were blocked with 5% skim milk at 37 °C for 1 h, incubated with primary antibodies at 4 °C overnight, and then incubated with the corresponding secondary antibodies at 37 °C for 2 h. Protein bands were detected using Ultra ECL-HRP Substrate (TU-ECL02, Tools Biotech, New Taipei City, Taiwan) and quantified using ImageJ 1.8.0 (National Institutes of Health, Bethesda, MD, USA).

### 2.7. Statistical Analysis

Results were presented as the mean ± standard error of the mean. The Mann–Whitney U test was used to compare differences between two independent groups. SPSS software version 18.0 (IBM, Armonk, NY, USA) was used for statistical analysis. Differences were considered statistically significant at *p* < 0.05.

## 3. Results

### 3.1. Bromelain Decreases Thrombus Formation in HFD-Fed Mice

Since PT, APTT, and FIB can characterize the intrinsic, extrinsic, and common pathways as well as the time of fibrin formation, these assays are commonly used to assess coagulation [19]. Plasma samples were used to detect PT, APTT, and FIB, and were found to be shortened in HFD-fed mice (Figure 1A) as compared to that in chow diet-fed mice, suggesting that clots could be formed in HFD-fed mice because of faster activation of the coagulation system. PT, APTT, and FIB were considerably prolonged in HFD-fed mice after treatment with bromelain (Figure 1B), indicating that bromelain can delay activation of the extrinsic, intrinsic, and common pathways by HFD. Additionally, the effect of bromelain on thrombosis of the liver was also investigated. H&E staining showed that daily administration of bromelain in for 12 weeks markedly decreased thrombus formation in the liver of HFD-fed mice (Figure 1C).

### 3.2. Effects of Bromelain on the Coagulation Cascade

Previous studies have reported that bromelain exerts anticoagulant effects; however, the detailed mechanism remains unclear [5]. Results of LC-MS/MS analysis showed that treatment with bromelain considerably affected eukaryotic initiation factor 2 signaling, acute phase response signaling, the coagulation system, including the intrinsic and extrinsic pathways, eIF4 and p70S6K signaling, and mTOR signaling (Figure 2). Compared to the HFD-fed group, treatment with bromelain increased the expression of SERPING1 and ATIII by over two-fold, suggesting that both the intrinsic and extrinsic pathways were inhibited (Table 1). Other proteins involved in the extrinsic pathway, such as tissue factor pathway inhibitor and vitamin K epoxide reductase complex subunit 1, were not affected by bromelain treatment. Additionally, bromelain treatment increased the expression of α1-antitrypsin and ATIII, suggesting that the common pathway was affected and, consequently, blood coagulation (Table 1). Apart from affecting the intrinsic, extrinsic, and common pathways, treatment with bromelain also affected the fibrinolytic system. Plasminogen levels were enhanced by bromelain treatment (Table 2). According to LC-MS/MS analysis, bromelain could suppress the intrinsic, extrinsic, and common pathways, and induce activation of the fibrinolytic system to inhibit blood coagulation. However, given that LC-MS/MS analysis is based on amino acid sequencing, it is necessary to verify changes in the expression of these proteins by Western blot assay.

### 3.3. Bromelain Increases Expression of ATIII, SERPING1, α1-Antitrypsin, Plasminogen, and Decreases Expression of FIB in the Liver and Plasma

To further confirm the mechanism by which bromelain inhibits coagulation, the protein expression of ATIII, SERPING1, α1-antitrypsin, plasminogen factor XIII, and FIB in the liver and plasma was investigated by Western blot assay. ATIII, SERPING1, and α1-antitrypsin play important roles in the suppression of coagulation [20,21,22]. We demonstrated that treatment with bromelain significantly upregulated the protein expression of ATIII, SERPING1, α1-antitrypsin, and plasminogen in the liver. Furthermore, bromelain also reduced the protein expression of FIB α and γ and factor XIII in the liver but did not affect the expression of FIB β (Figure 3A,B). The same effects were also observed in the plasma (Figure 4A–G).

## 4. Discussion

In the present study, we provide evidence to support the anticoagulant effects of bromelain and elucidate potential molecular mechanisms. To examine the effect of bromelain on NAFLD-induced thrombus formation, HFD-fed mice were used as an animal model for NAFLD pathology and deregulation of the coagulation system. We found that mice fed with HFD for 12 weeks induced hypercoagulation in the plasma and thrombosis in the liver, which is consistent with observations in human patients [15,18,19,23,24]. Treatment with bromelain attenuated the deregulation of blood coagulation, as evident by the results of the PT, APTT, and FIB time and thrombus formation in the liver, suggesting that bromelain exerts anticoagulant effects against NAFLD-induced deregulation of coagulation cascade activation. Since coagulation factors are mainly produced in the liver and primarily perform their physiological functions as proteins [17,18], we used LC-MS/MS analysis to assess their protein expression patterns instead of detecting coagulation factor mRNA expression. The results of LC-MS/MS analysis and coagulation factor activity suggest that bromelain-induced changes in PT, APTT, and FIB time may be attributed to altered protein expression of coagulation factors in the liver.

Previous studies have suggested that the mechanisms of the anticoagulant effects of bromelain may result from its proteolytic activity that promotes fibrin cleavage or disturbance in the activation of coagulation pathways [25,26,27]. However, the exact molecular mechanisms by which bromelain interferes with blood coagulation are unknown. In the present study, we demonstrated that bromelain increased the levels of anticoagulant proteins, including ATIII, SERPING1, α1-antitrypsin, and plasminogen, in the blood and liver of mice with HFD-induced NAFLD. ATIII is a well-known negative inhibitor of the intrinsic pathway and regulates the activity of factors IX, X, XI, and XII [20]. In addition, ATIII negatively regulates the extrinsic pathway by inhibiting the activity of factor VII [28]. Moreover, ATIII negatively regulates the common pathway by inhibiting thrombin synthesis [20]. Deletion of ATIII in mice leads to the death of embryos due to extensive thrombosis in the heart and liver [29], suggesting the crucial role of ATIII in regulating blood coagulation. On the other hand, SERPING1 is an inhibitor of factor XII [21,30,31] and mutations in SERPING1 cause hereditary angioedema [32,33]. Moreover, α1-antitrypsin is an inhibitor of thrombin and inhibits the common pathway, leading to the inhibition of clot formation [22,30]. In light of their functions, the findings of the present study strongly suggest that the increased levels of ATIII, SERPING1, and α1-antitrypsin may contribute to the beneficial effects of bromelain on NAFLD-induced deregulation of coagulation and clot formation.

When the coagulation system is activated, cleavage of FIB by thrombin results in fibrin clot formation and prevents bleeding [34,35]. In the present study, apart from upregulation of anticoagulant proteins, downregulation of FIB protein in the liver and plasma of HFD-fed mice was also observed, suggesting that bromelain may inhibit the production of FIB in the liver, and decreased FIB in the plasma may cause a decrease in the conversion of FIB to fibrin. Our results demonstrated that bromelain significantly decreased the protein expression of FIB α and γ chains in the liver; however, it did not affect the protein expression of the FIB β chains in the liver, which supports our findings that treatment with bromelain decreased the circulating levels of FIB. Moreover, we found that treatment with bromelain decreased the protein expression of factor XIII in the liver and plasma, which could lead to thrombus instability, making them prone to degradation. Although the exact molecular mechanisms remain to be elucidated, our findings suggest that the beneficial effects of bromelain on coagulation can be attributed to inhibition of the intrinsic, extrinsic, and common pathways.

Next, we investigated the effects of bromelain on the fibrinolytic system. To prevent thrombus formation after activation of the coagulation system, the fibrinolytic system is activated and converts plasminogen into plasmin, which degrades fibrin [35]. Under normal conditions, plasminogen is activated by tPA and uPA. Conversely, tPA and uPA are inhibited by PAI-1 [35,36,37]. Therefore, alterations in the expression of these proteins may induce thrombus formation. In the present study, treatment of HFD-fed mice with bromelain resulted in upregulation of plasminogen protein in the liver and plasma; however, no effect on the protein levels of tPA, uPA, and PAI-1 (data not shown). These results suggest that bromelain may increase fibrinolysis by enhancing the production of plasminogen in the liver and release in the plasma and consequently inducing the activation of the plasminogen–plasmin pathway, leading to a decrease in thrombus formation in the liver of HFD-fed mice. Collectively, these findings suggest that activation of the plasminogen-plasmin pathway is one of the regulatory mechanisms underlying the beneficial effects of bromelain on the NAFLD-induced deregulation of the coagulation system.

## 5. Conclusions

Our study provides insight into unique mechanisms through which bromelain regulates blood coagulation, including upregulation of ATIII, SERPING1, and α1-antitrypsin and downregulation of FIB and factor XIII in the liver and plasma, leading to inhibition of the intrinsic, extrinsic, and common pathways, as well as decreased clot stability. In addition, bromelain promotes the fibrinolytic system by activating plasminogen-to plasmin, thereby degrading clots, and ultimately leading to mitigation of NAFLD-induced thrombus formation (Figure 5). Taken together, the findings of the present study provide new insights into the beneficial effects of bromelain on dysregulation of blood coagulation induced by NAFLD, which broadens the biomedical implications of bromelain in the prevention or treatment of thrombosis-related disorders.

## Figures and Tables

**Figure 1 nutrients-14-02329-f001:**
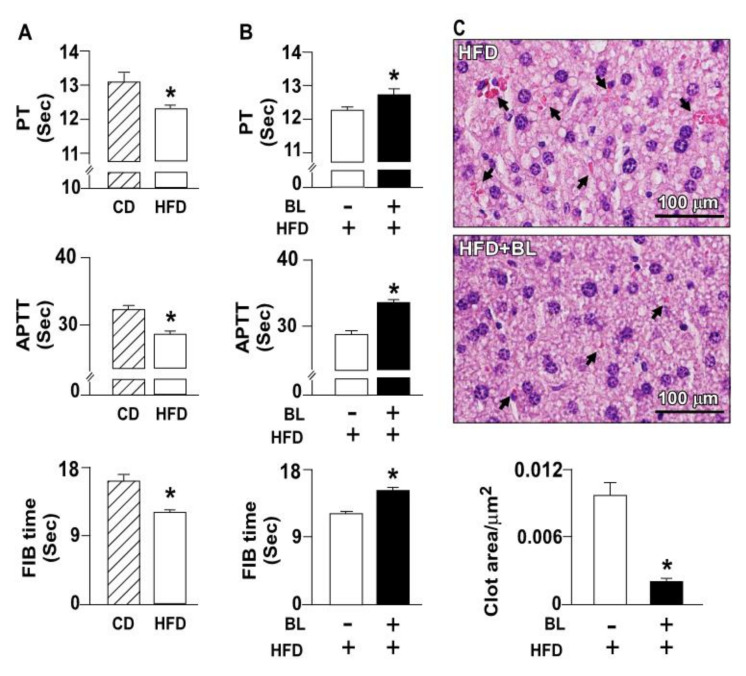
Bromelain (BL) alleviates high-fed diet (HFD)-induced deregulation of blood coagulation and decreases thrombus formation in the liver. (**A**) Prothrombin time (PT), activated partial thromboplastin time (APTT), and fibrinogen (FIB) time in chow diet (CD)- and HFD-fed mice. (**B**) PT, APTT, and FIB time in HFD-fed mice with or without BL administration. (**C**) Histological images of hematoxylin and eosin Y-stained liver samples and the quantification of clot area. Clots are indicated by arrows. Scale bar represents 100 μm. Data represent the mean ± standard error of mean (SEM) from 10 mice. * *p* < 0.05 vs. CD group or HFD alone group. Abbreviations: APTT, activated partial thromboplastin time; BL, bromelain; CD, chow diet; FIB, fibrinogen; HFD, high-fed diet; PT, prothrombin time.

**Figure 2 nutrients-14-02329-f002:**
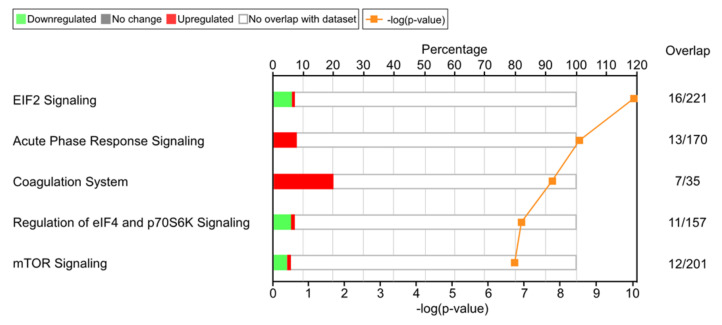
The top five signaling pathways affected by BL in the liver of HFD-fed mice. The downregulated or upregulated proteins were shown in green and red bars. The overlap numbers show how many altered proteins were in specific pathways. Abbreviations: EIF2, eukaryotic initiation factor 2; eIF4, eukaryotic translation initiation factor 4; mTOR, mammalian target of rapamycin; p70S6K, 70-kDa ribosomal protein S6 kinase.

**Figure 3 nutrients-14-02329-f003:**
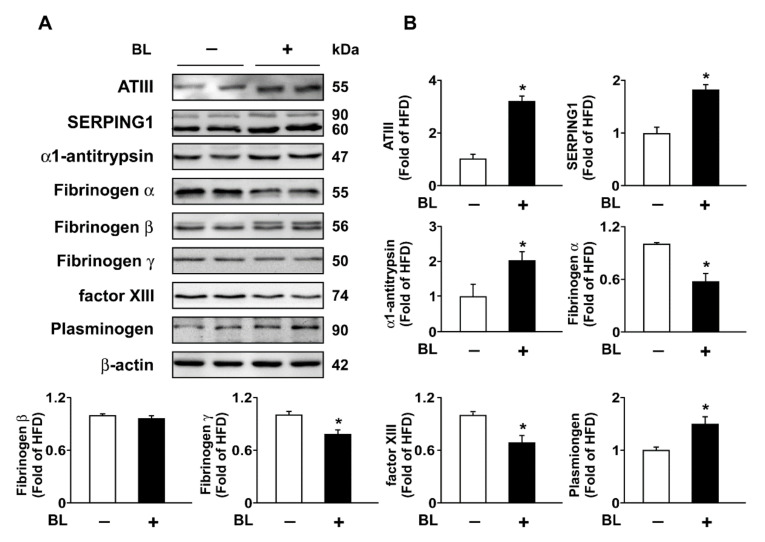
Effect of BL on the expression of coagulation factors in the liver of HFD-fed mice. Eight-week-old C57BL/6 mice were fed with HFD and administered phosphate-buffered saline (PBS) or BL (20 mg/kg/day) for 12 weeks. The livers were collected and subjected to Western blot analysis. (**A**,**B**) The protein levels of antithrombin III (ATIII), serpin family G member 1 (SERPING1), α1-antitrypsin, FIB α, FIB β, FIB γ, plasminogen, and β-actin. Data represent the mean ± SEM from 10 mice. * *p* < 0.05 vs. HFD alone group. Abbreviations: ATIII, antithrombin III; BL, bromelain; HFD, high-fed diet; SERPING1, serpin family G member 1.

**Figure 4 nutrients-14-02329-f004:**
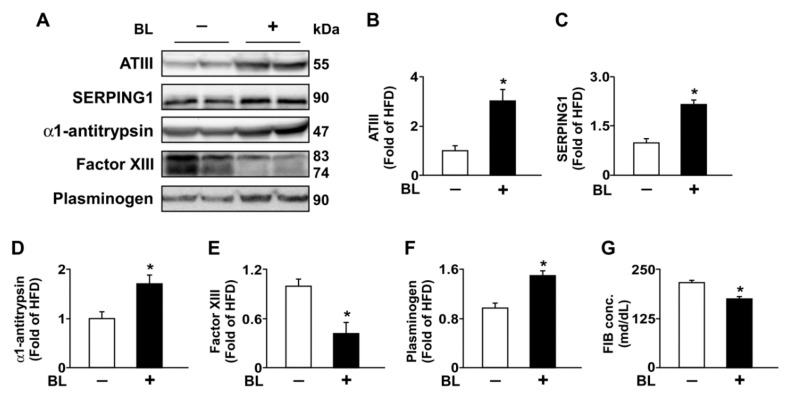
Effect of BL on the expression of coagulation factors in the plasma of HFD-fed mice. Eight-week-old C57BL/6 mice were fed with HFD and administered PBS or BL (20 mg/kg/day) for 12 weeks. The plasma was collected and subjected to Western blot analysis or enzyme-linked immunoassay (ELISA) assay. (**A**–**F**) The protein levels of ATIII, SERPING1, α1-antitrypsin, and plasminogen. (**G**) The levels of FIB were assessed by an ELISA kit. Data represent the mean ± SEM from 10 mice. * *p* < 0.05 vs. HFD alone group. Abbreviations: ATIII, antithrombin III; BL, bromelain; HFD, high-fed diet; SERPING1, serpin family G member 1.

**Figure 5 nutrients-14-02329-f005:**
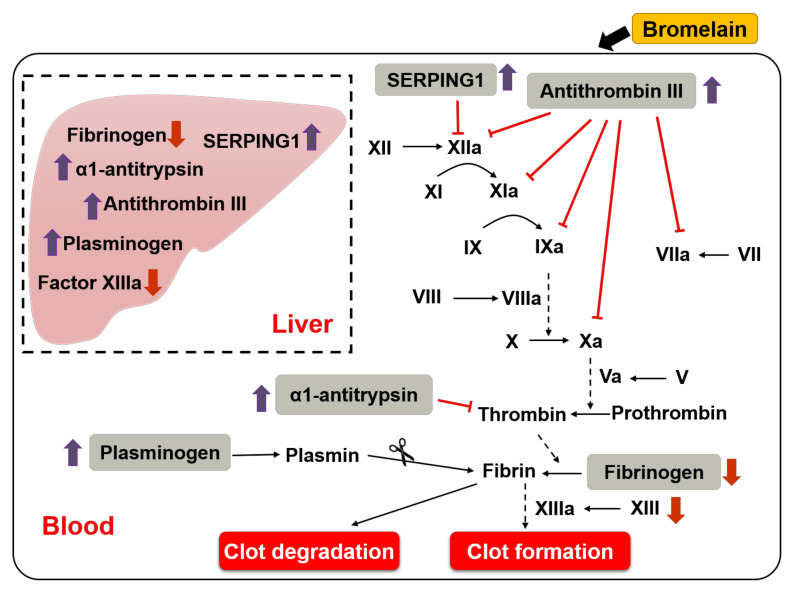
Schematic illustration of the molecular mechanism through which bromelain induces anticoagulant effects in HFD−fed mice. As shown, daily administration of BL for 12 weeks increased the expression of ATIII, SERPING1, α1-antitrypsin, plasminogen, and decreased the expression of FIB and factor XIII in the liver and plasma of HFD-fed mice, leading to inhibition of the intrinsic, extrinsic, and common pathways and activation of fibrinolytic pathway. These events may work in concert to inhibit NAFLD-induced thrombus formation. Abbreviations: SERPING1, serpin family G member 1. Red and purple arrows mean upregulation and downregulation.

**Table 1 nutrients-14-02329-t001:** Effect of bromelain on the expression of proteins involved in the intrinsic, extrinsic, and common pathways in the liver of HFD−fed mice.

Protein	HFD + BL vs. HFD	Regulation of Coagulation Activity	Intrinsic	Extrinsic	Common
SERPING1	↑	Negative	✓		
Antithrombin III	↑	Negative	✓	✓	✓
Annexin A5	−	Negative	✓		✓
Plasma kallikrein	−	Positive/Negative	✓		
Tissue factor pathway inhibitor	−	Negative	✓	✓	
Vitamin K epoxide reductase complex subunit 1	−	Positive	✓	✓	
α1-antitrypsin	↑	Negative			✓
Prothrombin	−	Positive			✓

↑, upregulated. Abbreviations: BL, bromelain; HFD, high-fed diet; SERPING1, serpin family G member 1.

**Table 2 nutrients-14-02329-t002:** Effect of bromelain on the expression of proteins involved in the fibrinolytic system in HFD−fed mice.

Protein	HFD + BL vs. HFD	Regulation of Coagulation Activity
Plasminogen	↑	Negative
α2-antiplasmin	−	Positive
Plasma kallikrein	−	Positive/Negative
Plasminogen activator inhibitor-1	−	Positive
Tissue plasminogen activator	−	Negative
Urokinase-type plasminogen activator	−	Negative

↑, upregulated. Abbreviations: BL, bromelain; HFD, high-fed diet.

## Data Availability

Data supporting the findings of this study are available from the corresponding author upon reasonable request.

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
