# Peer review of "New Mechanisms of Bromelain in Alleviating Non-Alcoholic Fatty Liver Disease-Induced Deregulation of Blood Coagulation"

_nutrients, 2022, doi:10.3390/nu14112329_

Round 1

Reviewer 1 Report

A paper entitled “New mechanisms of bromelain in alleviating non-alcoholic 2 fatty liver disease-induced deregulation of blood coagulation” seems to be interesting. Authors shown that bromelain, a cysteine-rich protease found in pineapples, can exerts anticoagulant effects in NAFLD-induced mouse model of coagulation disturbance.

Introduction section is composed correct, giving prelude to the topic. Then methods section also remines described it in a factual and professional manner. However, I would suggest making a revision of results and discussion section:

1.      The results section contains many introductions and interpretations that should be transferred to the introduction or discussion section, e.g .:

-        Lines 158-163 – should be moved to introduction,

-        Lines 171-174 ; 185-186; 190-194; 201; 223-226– should be moved to discussion

2.      Discussion section is too long, what make it hard to follow. I would suggest rewriting it to highlight the most important and promising content.

3.      Figure 1(C) – please make images quantification – clots/µm2- and add a graph to this figure, otherwise authors should not discuss changes in clots formation in the liver after the treatment.

4.      Results 3.2. Effects of bromelain on the coagulation – a figure showing LC-MS/MS results is needed

5.      Figure 2. – I would suggest decreasing the size of this graph

6.      Figure 4(A) – very pore quality of representative WB membranes (especially for Factor XIII), please find better representation

7.      Authors did not present a result for mRNAs expression of studied factors (only proteins expression). The authors should either add mRNA expression results or refer to why it is not included in the publication. The fact that there is no information as to whether the displayed clotting factor protein levels were native or active is also a weak point. On the other hand, authors provided a PT, APTT and FIB time, so they may defense their thesis but more comment in discussion section about it is needed.

However, present paper is valuable source of information, and I would accept this paper after revision.   

Author Response

Reviewer #1

A paper entitled “New mechanisms of bromelain in alleviating non-alcoholic 2 fatty liver disease-induced deregulation of blood coagulation” seems to be interesting. Authors shown that bromelain, a cysteine-rich protease found in pineapples, can exerts anticoagulant effects in NAFLD-induced mouse model of coagulation disturbance.

Introduction section is composed correct, giving prelude to the topic. Then methods section also remines described it in a factual and professional manner. However, I would suggest making a revision of results and discussion section:

1. The results section contains many introductions and interpretations that should be transferred to the introduction or discussion section, e.g .:

- Lines 158-163 – should be moved to introduction,

Response: We fully agree with the reviewer’s viewpoint. In response to the reviewer’s suggestion, we have moved these sentences to introduction section in our revised manuscript.

- Lines 171-174 ; 185-186; 190-194; 201; 223-226– should be moved to discussion

Response: We fully agree with the reviewer’s viewpoint. In response to the reviewer’s suggestion, we have moved these sentences in our revised manuscript.

2. Discussion section is too long, what make it hard to follow. I would suggest rewriting it to highlight the most important and promising content.

Response: In response to the reviewer’s suggestion, we have shortened the section of discussion of our revised manuscript.

3. Figure 1(C) – please make images quantification – clots/µm2- and add a graph to this figure, otherwise authors should not discuss changes in clots formation in the liver after the treatment.

Response: We thank the reviewer’s professional suggestion. In response to the reviewer’s suggestion, we made image quantification of clot area in the liver section of NAFLD mice. We have reported these new data in the figure 1C in our revised manuscript. Accordingly, the figure legend has been also revised.

4. Results 3.2. Effects of bromelain on the coagulation – a figure showing LC-MS/MS results is needed

Response: We thank the reviewer’s professional suggestion. In response to the reviewer’s suggestion, we have provided LC-MS/MS results in the figure 2 in our revised manuscript. Accordingly, the figure legend has been also revised.

5. Figure 2. – I would suggest decreasing the size of this graph.

Response: We fully agree with the reviewer’s viewpoint. In response to the reviewer’s suggestion, we have provided new image of figure 2 to replace the original one in our revised manuscript.

6. Figure 4(A) – very pore quality of representative WB membranes (especially for Factor XIII), please find better representation

Response: We fully agree with the reviewer’s viewpoint. In response to the reviewer’s suggestion, we have provided new image of Factor XIII protein to replace the original one in our revised manuscript.

7. Authors did not present a result for mRNAs expression of studied factors (only proteins expression). The authors should either add mRNA expression results or refer to why it is not included in the publication. The fact that there is no information as to whether the displayed clotting factor protein levels were native or active is also a weak point. On the other hand, authors provided a PT, APTT and FIB time, so they may defense their thesis but more comment in discussion section about it is needed.

Response: We fully agree with the reviewer’s viewpoint. We have discussed this point in our revised manuscript. Now, the paragraph read as Treatment with bromelain attenuated the deregulation of blood coagulation as evident by the results of PT, APTT and FIB time and thrombus formation in the liver, suggesting that bromelain exerts anticoagulant effects against NAFLD-induced deregulation of coagulation cascade activation. Since coagulation factors are mainly produced in the liver and primarily perform their physiological functions as proteins [17,18], we used LC-MS/MS analysis to assess their protein expression patterns instead of detecting coagulation factor mRNA expression. The results of LC-MS/MS analysis and coagulation factor activity suggest that bromelain-induced changes in PT, APTT and FIB time may be attributed to altered protein expression of coagulation factors in the liver”.

We sincerely hope the reviewer can approve our response.

8. However, present paper is valuable source of information, and I would accept this paper after revision.

Response: We thank the reviewer for the positive feedback.

Reviewer 2 Report

The article entitled New mechanisms of bromelain in alleviating non-alcoholic fatty liver disease-induced deregulation of blood coagulation is very interesting.

I have no comments for the authors.

Author Response

The article entitled New mechanisms of bromelain in alleviating non-alcoholic fatty liver disease-induced deregulation of blood coagulation is very interesting. I have no comments for the authors.

Response: We thank the reviewer for the positive feedback.

Round 2

Reviewer 1 Report

I highly recommend a paper entitled “New mechanisms of bromelain in alleviating non-alcoholic 2 fatty liver disease-induced deregulation of blood coagulation” to be published in Nutrients in the present form.